



# Acute changes in macronutrient stoichiometry alter nitrate uptake in benthic biofilms

Anika Große[1], Alexander J. Reisinger[2], Nuria Perujo[3], Patrick Fink[1,3,4], Dietrich Borchardt[1], and Daniel Graeber[1]

[1]Helmholtz-Centre for Environmental Research - UFZ, Department Aquatic Ecosystem Analysis, Brückstr. 3a, 39114 Magdeburg, Germany
[2]Department of Soil, Water, and Ecosystem Sciences - University of Florida, 32601 Gainesville, FL, USA
[3]Helmholtz-Centre for Environmental Research - UFZ, Department River Ecology, Brückstr. 3a, 39114 Magdeburg, Germany
[4]University of Cologne, Institute of Zoology, Zülpicher Straße 47b, 50674 Köln, germany

**Correspondence:** Anika Große (anika.grosse@ufz.de)

**Abstract.** Benthic biofilms, located at the sediment-water interface, are hot-spots for macronutrient cycling in headwater streams. Here, the supply of dissolved organic carbon (DOC), nitrogen (N), and phosphorus (P), affects nutrient cycling processes such as nitrate uptake. Flushing events can add short-term pulses of DOC, N and P to streams, changing the macronutrient ratios in the stream water, altering stoichiometric imbalances between water and microbial macronutrient ratios. However, there is little information on whether these short-term changes in macronutrient imbalances can alter biofilm nitrate uptake. To better understand how acute changes to DOC, N, and P stoichiometric imbalances affect nitrate uptake, we sampled stream biofilms from four different sites in Florida and incubated them in the lab in mesocosms after changing their macronutrient ratios by adding DOC and/or nitrate. Here we show that biofilms from anthropogenically less impacted streams with less N excess increased their nitrate uptake after 48 h of incubation in different macronutrient stoichiometric ratios, but biofilm structure remained mainly unaffected. Furthermore, nitrate uptake was positively related to biofilm metabolism, differentiating in sites with more autotrophic- or more heterotrophic-driven nitrate uptake. Our study reveals that acute changes in macronutrient stoichiometric imbalance between stream water and biofilm microorganisms changes nitrate uptake. This needs to be considered when assessing short-term nitrate uptake capacity of stream reaches.

## 1 Introduction

High concentrations of inorganic nitrogen (N) in ground and surface waters negatively affect aquatic ecosystems through eutrophication and reduced drinking water quality, having negative environmental, economic, and human health consequences (Camargo and Alonso, 2006). Global anthropogenic N and phosphorus (P) inputs into surface waters have increased in the last century (Beusen et al., 2016), as have the molar N : P of these inputs into the biosphere over the past decades (Peñuelas et al., 2012). Nonpoint source inputs of N are difficult to control, particularly compared to point sources (Carpenter et al., 1998).

Stream biofilms are hotspots for biogeochemical processes like carbon (C) and N cycling, organic matter decomposition, respiration and primary production, especially in headwaters (Battin et al., 2016). Therefore, biofilms have the potential to





affect water quality in downstream ecosystems (Alexander et al., 2007; Peterson et al., 2001). Benthic biofilms are located at the sediment water interface and are composed of algae, bacteria, fungi, protozoa and meiofauna embedded in a matrix of extracellular polymeric substances (EPS; Battin et al., 2016), that can store organic matter to buffer for macronutrient changes in the water column (Freeman and Lock, 1995). The most reactive zone of stream sediments are the upper 2 cm, also known as the benthic biolayer (Knapp et al., 2017). Both structural (e.g., bacterial density, chlorophyll a (Chl a) concentration) and functional (e.g., community-levl physiological profiles (CLPPs) or bacterial production) variables representing benthic biofilms show a strong reactivity to changes in macronutrient ratios (Große et al., 2025a).

As biogeochemical cycles in aquatic ecosystems are closely linked, the uptake of one macronutrient can depend on the sufficient availability of others (Schade et al., 2011; Große et al., 2025b). Nitrate-N ($NO_3-N$) can be removed from the water column via assimilatory uptake into biomass or by dissimilatory processes such as denitrification, which leads to a permanent removal of N from the system as $N_2$ gas (Mulholland et al., 2008). Assimilatory uptake of $NO_3-N$ is an energy-demanding process that requires organic C as an energy source for heterotrophs, or light for autotrophs (Mulholland et al., 2006; Rodríguez-Cardona et al., 2021). Under low-oxygen conditions, denitrification uses $NO_3$ as an electron acceptor to obtain energy from organic C. Denitrification and assimilation of $NO_3-N$ therefore depend on the availability of dissolved organic carbon (DOC), particularly on the molar ratio of $DOC : NO_3-N$. This stoichiometric perspective considers not only the concentration of individual macronutrients, but also the balance between them which can help identify the nutrient most likely to limit specific ecological processes (Sterner and Elser, 2002). The $DOC : NO_3-N$, in particular, can indicate stoichiometric limitations on $NO_3-N$ uptake, making it a key factor in explaining $NO_3-N$ accumulation in ecosystems worldwide (Taylor and Townsend, 2010). In line with this, the ratio of $DOC : NO_3-N$ has been shown to be a better predictor for $NO_3-N$ uptake velocities than the $NO_3-N$ concentration (Rodríguez-Cardona et al., 2016). Other studies have linked $NO_3-N$ uptake to the DOC : dissolved inorganic N (DIN) : soluble reactive P (SRP) in various settings, including vertical flow through columns (Stutter et al., 2020), stream water microcosms (Graeber et al., 2021; Peñarroya et al., 2024) or stream mesocosms (Große et al., 2025b). For these ratios, it is crucial to consider the bioavailable fraction of DOC, the reactive DOC (rDOC), because heterotrophic $NO_3-N$ uptake can still be limited by DOC availability, even when total DOC concentrations are relatively high (Graeber et al., 2021; Pasqualini et al., 2024; Peñarroya et al., 2024). The relevance of rDOC is underscored by a recent study showing that adding DOC and SRP to shift conditions from DIN excess toward a rDOC : DIN : SRP closer to the Redfield ratio significantly enhanced $NO_3-N$ uptake, even though $NO_3-N$ remained in excess (Große et al., 2025b). The aforementioned studies assessed effects of changed imbalances largely for streams under stoichiometric $NO_3-N$ excess, while studies for steams under N depletion remain scarce. Thus, a deeper understanding of the $NO_3-N$ uptake in stream biofilms requires further research under conditions with balanced rDOC : DIN : SRP or under N depleted conditions.

The connection between the rDOC : DIN : SRP and macronutrient uptake can be considered through the framework of the "macronutrient access hypothesis", which was developed for planktonic bacteria in stream water by Graeber et al. (2021). This hypothesis proposes that nutrient assimilation by aquatic heterotrophs is governed by the balance between their stoichiometric macronutrient demands and the macronutrients that are actually accessible to them. Applied to $NO_3-N$, the highest relative uptake can be expected under N-limitation at high rDOC : DIN and low DIN : SRP, compared to microbial demand C : N : P.





While nutrient ratios influence biofilm $NO_3-N$ uptake, the underlying metabolic processes within these complex communities may also play a key regulatory role. Autotrophic and heterotrophic contributions to $NO_3-N$ dynamics can be distinguished by estimating aerobic metabolic activity (Wachholz et al., 2024). $NO_3-N$ uptake in stream ecosystems has been correlated

to light availability (Wymore et al., 2016), canopy cover (Beaulieu et al., 2014; Reisinger et al., 2019) and gross primary production (GPP; Hall Jr. et al., 2009; Hall and Tank, 2003; Mulholland et al., 2008) and were therefore connected to autotrophs for modeling $NO_3-N$ uptake in streams (Rode et al., 2016). Heterotrophic contributions to $NO_3-N$ uptake have been shown as a correlation of ecosystem respiration and denitrification (Mulholland et al., 2008) or a correlation of $NO_3-N$ uptake with ecosystem respiration (Beaulieu et al., 2014). Furthermore, the products of photosynthesis can be used by heterotrophs as

organic C substrates for denitrification (Heffernan and Cohen, 2010). As of yet, the links between metabolic processes and stoichiometric imbalances of macronutrients remain unclear due to a lack of studies, combining these two perspectives.

For stream biofilms, it is necessary to distinguish between chronic macronutrient pollution and pulsed, acute additions, as short-term nutrient additions elicit a different functional response than plateau additions (Weigelhofer et al., 2018). Short-term macronutrient pulses and changes in stream water DOC : DIN : total dissolved P can be connected to high flow events

through agricultural drainage (Smith and Jarvie, 2018), or transport of DOC and $NO_3-N$ from the soil of surrounding riparian forests (Ledesma et al., 2022). In some catchments, the majority of DOC transported downstream enters the stream as pulses (Graeber et al., 2018). In another case, hurricanes in urban catchments export pulses of bioavailable DOC in combination with inorganic macronutrients and have the potential to change structure and function in the receiving stream ecosystems (Chen et al., 2019). Field data show that $NO_3-N$ assimilation and denitrification are interrupted during larger and moderate storm

events, leading to a larger downstream transport (Hill, 2023). In contrast, Bernal et al. (2019) describe for storm events a similar or higher demand for DOC and $NO_3-N$ compared to base flow. However, nutrient pulses are not limited to storm events. Snow melt, rewetting after drought, groundwater movements, drainage utilization or biological seasonal processes also affect macronutrients in streams without flood events (McAleer et al., 2022; Sebestyen et al., 2008; Winter et al., 2023). The relationship between the uptake of macronutrients by heterotrophs and autotrophs in streams can change within hours

(Johnson et al., 2012; Liu et al., 2015; Schade et al., 2011), even if the biofilm structure remains unaffected (Besemer, 2015). Nevertheless, effects of short-term changes of macronutrient ratios on biofilm structure and function remain unclear.

To better understand the effect of short term changes in macronutrient stoichiometry on benthic biofilm structure, functioning, and $NO_3-N$ uptake, we sampled benthic stream sediments from four sites in three streams (one stream was sampled at two separate locations) with different watershed land use in Florida. The four sites differed in their background macronutrient stoi-

chiometry and we manipulated the water DOC : DIN : phosphate-P ($PO_4-P$) for 48 h in a microcosm experiment to simulate synchronous (DOC and $NO_3-N$ addition together) and asynchronous pulses (DOC or $NO_3-N$ addition). We hypothesized (I) that the short timeframe and the pulse nature of the nutrient addition in our experiment should favor functional biofilm responses, as we postulate a faster reaction by functional rather than structural biofilm variables. These functional variables include $NO_3-N$ uptake and metabolism as key functions in biogeochemical cycling. Following the "macronutrient access

hypothesis", we hypothesized (II) that the highest relative $NO_3-N$ uptake rates would be found in benthic biofilms grown in sites with N limited conditions, defined by high DOC : DIN and low DIN : $PO_4-P$ in the water column. We expect a combi-




nation of heterotrophic and autotrophic assimilation and therefore hypothesized (III) a positive effect of GPP and community respiration (CR) on $NO_3-N$ uptake.

## 2 Material and methods

### 2.1 Site description

The experiment was conducted using sediment samples collected from streams in Gainesville, Florida, USA and the surrounding area. In general, Florida's geology is characterized by phosphate-containing rocks (USGS, 1963), which lead to high background concentrations of P in aquatic ecosystems. This elevated P is particularly true in this region of Florida, where the P-rich Hawthorne Formation contributes to naturally elevated P in surface waters (Scott et al., 1983). We selected four sites with differing land use in their catchment along a gradient of anthropogenic impact. The first sampling site, Hatchet Creek, was located in a forested watershed maintained primarily as conservation lands and represents the most natural sampling site. The second and third sampling sites were both located on Hogtown Creek, with a more upstream and a more downstream site. The upstream Hogtown Creek site drained a primarily residential watershed, and the downstream site flowed through conservation lands within the city just upstream of our sampling location. The most anthropogenically impacted sampling site, Sweetwater Branch, was located downstream of a wastewater treatment plant and drains much of downtown Gainesville. The different land uses of each catchment can be found in Table 1.

We used land use data to estimate the background bioavailability of DOC, which was then used to calculate rDOC. We multiplied the bioavailability data from Stutter et al. (2018) with the share of each land use in the catchments from Table 1, resulting in DOC bioavailability estimates of 9.6 % at Hatchet Creek, 14.0 % at Hogtown Creek (upstream), 15.0 % at Hogtown Creek (downstream) and 16.3 % at Sweetwater Branch.

### 2.2 Experimental setup

The experiment was conducted in November 2023. A small measuring spoon was used to consistently collect and transfer approximately 50 cm$^3$ of (the predominantly sandy) sediment from the streambed into a 250 mL glass mason jar, followed by the addition of 150 mL of stream water. An additional sediment sample was collected for background biofilm isotope ratios R($^{15}$N/$^{14}$N) and stored frozen until later processing. The mason jars were transported to the laboratory in the dark and subsequently placed under a grow lamp with photosynthetically active radiation (PAR) of 784 $\mu$mol photons m$^{-2}$ s$^{-1}$ to reach light saturated conditions. The jars were kept under a 12 h light : 12 h dark photoperiod without lids for the duration of the experiment.

Following a 24 h acclimation period, the water in the mason jars was gently decanted and replaced with 150 mL of fresh site water to ensure sufficient oxygen and nutrient availability. At this point, macronutrient treatments were initiated using a full-factorial design. One factor involved an increase in DOC concentration by 7 mg C L$^{-1}$ through the addition of sodium acetate; the second factor involved an increase in $NO_3-N$ concentration by 0.5 mg N L$^{-1}$ through the addition of sodium nitrate. This



**Table 1.** Coordinates and catchment characteristics of sampling sites (Taylor, 2023)

| Parameter | Unit | Hatchet Creek | Hogtown Creek (upstream) | Hogtown Creek (downstream) | Sweetwater Branch |
|---|---|---|---|---|---|
| Longitude | | -82.21 | -82.35 | -82.39 | -82.32 |
| Latitude | | 29.69 | 29.67 | 29.64 | 29.63 |
| Mean PAR | $\mu$mol photons m$^{-2}$ s$^{-1}$ | 16.43 | 61.76 | 75.34 | 75.90 |
| Total area | km$^2$ | 88 | 16 | 51 | 5.8 |
| Impervious surface cover | % | 2 | 22 | 17 | 29 |
| Road density | km km$^{-2}$ | 1.2 | 6.8 | 8 | 13 |
| Flashiness rank | | 1.5 | 4 | 2 | 5 |
| Urban and built-up | % | 9 | 66 | 75 | 83 |
| City infrastructure | % | 1 | 5 | 4 | 9 |
| Upland forest | % | 58 | 15 | 8 | 4 |
| Wetlands | % | 25 | 9 | 13 | 4 |
| Open land | % | 1 | 0 | 0 | 0 |
| Agriculture | % | 6 | 0 | 1 | 0 |

resulted in four treatments: control (cn) with no additions, elevated DOC and background $NO_3-N$ (Cn), background DOC and elevated $NO_3-N$ (cN), and elevated DOC and $NO_3-N$ (CN). The combination of DOC and $NO_3-N$ additions allowed us to

cover a broad range of DOC : DIN : $PO_4-P$ (Fig. 3a).

Sediment samples were allowed to acclimate for an additional 24 h after treatment initiation. Subsequently, the water was replaced again and DOC and $NO_3-N$ were re-applied according to the respective treatments. Metabolic measurements (GPP and CR) were then performed (see below), followed by another water exchange and the addition of a stable isotope tracer to measure $NO_3-N$ in the biofilms (see method description below). Another 24 h after the addition of the stable isotope tracer,

water and sediment samples were collected as detailed in the following methods section. The timeline of the experiment is illustrated in Fig. 1.

For the addition of a $^{15}N$ stable isotope tracer ($K^{15}NO_3$), we followed the methods described in Mulholland (2004) and Tank et al. (2017). We assumed the background $^{15}N/^{14}N$ ratios for "rangeland and forest" for Hatchet Creek and "urban" for both sites at Hogtown Creek and Sweetwater Branch from Phelbs (2004) and increased the $^{15}N/^{14}N$ ratio by 5000. This approach

led to an increase in $NO_3-N$ of 7 % in Hatchet Creek and 13 % in both sites at Hogtown Creek and Sweetwater Branch.





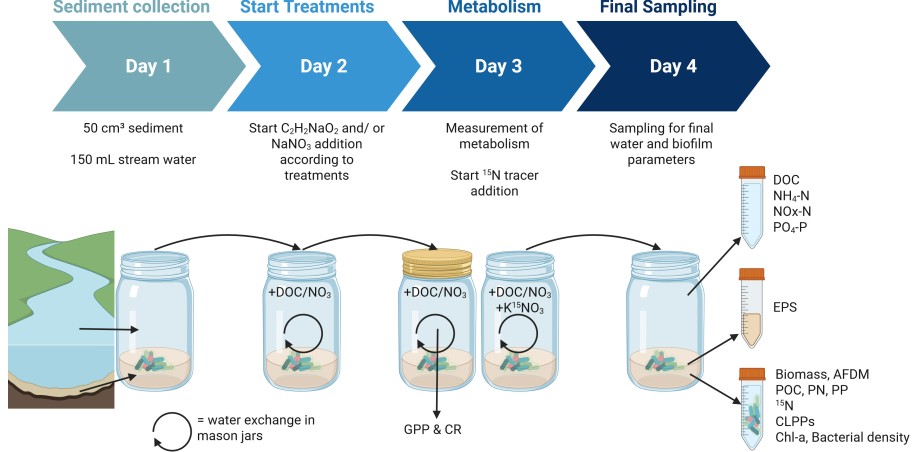

**Figure 1.** Experimental setup including timeline, main steps and measured water and biofilm parameters. Created in BioRender. Große, A. (2025) https://BioRender.com/9jvky31

## 2.3 Water physical and chemical parameters

### 2.3.1 Incubation startpoint

We measured physical and chemical parameters in situ when sediments were collected from the streams using a YSI-ProDSS (YSI Inc., Yellow Springs, OH, USA) that was calibrated before and after field work according to manufacturer recommenda-
tions (Table 2). Background water chemistry samples were filtered through 0.22 $\mu$m (nominal pore size) polyethersulfone filters (MilliporeSigma, Burlington, MA, USA) in the field into sterile scintillation vials (for N and P species) or acid-washed brown HDPE Nalgene bottles (DOC, total dissolved N). DOC (limit of quantification (LOQ) = 0.004 mg L$^{-1}$) and total dissolved N concentrations (LOQ = 0.005 mg L$^{-1}$) were measured using a Shimadzu TOC-L total organic carbon analyzer equipped with a TNM-L total nitrogen module (Shimadzu Corp., Kyoto, Japan). Concentrations of $NO_3-N$, ammonium-N ($NH_4$-N), and
$PO_4-P$ were determined according to the corresponding EPA standard methods (EPA 353.2, 350.1, and 365.1), with LOQ of 0.5 mg L$^{-1}$ for N species and 0.01 $\mu$g L$^{-1}$ for $PO_4-P$.

### 2.3.2 Incubation endpoints

Water samples were collected from mason jars at the end of the experiment (day 3) to quantify the remaining DOC, $NH_4$-N, $NO_X-N$ (sum of $NO_3-N$ and nitrite-N) and $PO_4-P$. We measured $NH_4$-N with the phenol-hypochlorite method (LOQ = 0.011
mg L$^{-1}$), $NO_X-N$ with the cadmium reduction method (LOQ = 0.012 mg L$^{-1}$) and $PO_4-P$ with the ascorbic acid method (LOQ = 0.005 mg L$^{-1}$) on an AQ400 Discrete Analyzer (Seal Analytical, Mequon, WI). DOC and TDN were analyzed the same way as the stream water samples described above. The calculation of the initial rDOC was based on the background rDOC of the stream water and added the increase of 7 mg L$^{-1}$ for the DOC addition treatments. The initial $NO_X-N$ concentra-



**Table 2.** Physicochemical parameters measured during sediment sampling, sampling sites and days.

| Parameter | Unit | Hatchet Creek | Hogtown Creek (upstream) | Hogtown Creek (downstream) | Sweetwater Branch |
|---|---|---|---|---|---|
| Sampling Date | | 13 November 2023 | 28 November 2023 | 27 November 2023 | 14 November 2023 |
| pH | | 6.43 | 7.56 | 6.34 | 7.2 |
| Turbidity | NTU | 3.07 | 1.44 | 3.88 | 1.45 |
| Conductivity | $\mu$S cm$^{-1}$ | 152.9 | 236.9 | 186.6 | 755 |
| Temperature | °C | 18.4 | 14.2 | 16.5 | 22.0 |
| Oxygen saturation | % | 39.8 | 93.5 | 44.6 | 92.7 |
| Oxygen concentration | mg L$^{-1}$ | 3.74 | 9.59 | 4.33 | 8.1 |

tion of treatments with N addition was calculated as the sum of the stream water $NO_X-N$ concentration and the experimental 0.5 mg N L$^{-1}$ addition. To calculate the change in DOC ($\Delta$ DOC), NH$_4$-N ($\Delta$ NH$_4$-N), $NO_X-N$ ($\Delta$ $NO_X-N$) and $PO_4-P$ ($\Delta$ $PO_4-P$) concentrations during the incubations, the measured concentrations in each mason jars at the end of the experiment were subtracted from the start concentrations.

## 2.4 Biofilm parameters

Both GPP and CR were measured as indicators of metabolic activity in each mason jar on day 3 of the experiment. To this end, changes in dissolved oxygen concentrations were recorded under both light and dark conditions following Gallagher and Reisinger (2020). For each incubation, fresh water was used, and mason jars were completely filled to eliminate air bubbles and ensure airtight conditions. Net primary production (NPP) and CR were calculated as changes in dissolved oxygen during light and dark incubation, respectively, and GPP was calculated as the sum of NPP and the absolute value of CR.

Biofilm samples were collected on day 4, which represented the end of the experiment. For EPS analysis, a sediment sub-
sample of approximately 3 cm$^3$ was collected following the method described by DuBois et al. (1956), with modifications according to Perujo et al. (2017). To extract the biofilm for the $^{15}$N measurement, an additional sediment subsample of approximately 15 cm$^3$ was transferred into a 50 mL centrifuge tube containing 20 mL of ultrapure water. Biofilm detachment was achieved through vortexing for 5 s, followed by 60 s in an ultrasonic bath, and a subsequent 5 s vortexing step. The resulting supernatant was transferred to a new centrifuge tube, freeze-dried (Delta 2–24 LSCplus, Germany), and analyzed for
$^{15}$N content using an elemental analyzer (Flash HT 2000, Thermo Fisher Scientific, Germany) coupled to an isotope ratio mass spectrometer (Delta V Advantage IRMS, Thermo Fisher Scientific, Germany).

Calculations of $NO_3-N$ uptake rates followed the approach outlined by Pasqualini et al. (2024), with adaptations to express uptake in mg N per gram of sediment (sed). Initially, the measured isotope ratios R($^{15}$N/$^{14}$N) of enriched and background samples were converted into the excess atom fraction xE($^{15}$N)$_{PN}$, defined as the molar fraction $^{15}$N/($^{14}$N+$^{15}$N), following the



LINX II Stream $^{15}$N Experiment Protocols (Mulholland, 2004). The amount of $^{15}$N incorporated into the sediment [mg $^{15}$N $g_{sed}^{-1}$] was then calculated using the following equation:

$$^{15}N_{sed} = PN \times xE(^{15}N)_{PN} \tag{1}$$

where PN represents particulate N in the sediment [mg N $g_{sed}^{-1}$] (method described below), and xE($^{15}$N)$_{PN}$ is the excess atom fraction of the $^{15}$N-labeled biofilm. Based on these data, we adapted the calculation for absolute $NO_3-N$ uptake rate

($U_{sed}$ in mg N $g_{sed}^{-1}$ $d^{-1}$) as:

$$U_{sed} = \frac{^{15}N_{sed}}{xE(^{15}N)_{H_2O}} \times 1d^{-1} \tag{2}$$

The excess atom fraction xE($^{15}$N)$_{H2O}$ is defined as the molar fraction $^{15}$N/($^{14}$N+$^{15}$N) of the mason jar water. Because the tracer addition lasted for 24 h, $NO_3-N$ uptake values were expressed as daily uptake rates without requiring additional temporal conversion. Relative $NO_3-N$ uptake ($U_{rel}$ expressed as a % of initial $NO_X-N$) was calculated as:

$$U_{rel} = \frac{U_{sed}}{NO_x - N_{initial}} \times 100\% \tag{3}$$

where $NO_X-N_{initial}$ is the initial $NO_X-N$ concentration in the mason jar.

For all other parameters, we detached the biofilm from the remaining sediment in the mason jars by mixing the sediment with 50 mL of Ringer's solution (Thermo Scientific, Waltham, USA) and subjecting this mixture to the same detachment procedure as described above: vortexing for 5 s, followed by 60 s in an ultrasonic bath, and a second 5 s vortex. The resulting

supernatant—hereafter referred to as biofilm suspension—was used for the analysis of particulate organic carbon (POC), particulate nitrogen (PN), particulate phosphorus (PP), Chl a, bacterial density, biomass, ash-free dry mass (AFDM), and CLPPs. To determine POC and PN, 2 mL aliquots of the biofilm suspension were filtered onto pre-combusted glass fiber filters (Cytiva Whatman GF/F filters, nominal pore size: 0.7 $\mu$m), dried at 60°C, and treated with diluted hydrochloric acid of ph 3 to remove inorganic carbon. The samples were analyzed via gas measurement following high-temperature combustion using a

vario EL cube elemental analyzer (Elementar Analysensysteme GmbH, Hanau, Germany). An additional 10 mL aliquot of the biofilm suspension was used for PP analysis. PP samples were acidified with 300 $\mu$L sulfuric acid (1 : 4 diluted), digested in a heating block, and analyzed photometrically according to DIN EN ISO 6878. C : N : P in initial and final mason jar water samples (rDOC : DIN : $PO_4-P$) and biofilm samples (POC : PN : PP) were visualized using a Redfield ratio-normalized ternary diagram (Graeber et al., 2021).

CLPPs and metabolic substrate use were assessed using Biolog EcoPlates© (Biolog Inc., Hayward, USA), which contain 31 carbon substrates and a tetrazolium dye that changes color upon microbial reduction. For this approach, 1 mL of the biofilm suspension was diluted with 15 mL of Ringer's solution, and 130 $\mu$L of the diluted suspension was inoculated into each well of the EcoPlate. Absorbance at a wavelength of 590 nm was measured using a microplate reader (Epoch 2, BioTek Instruments,



Winooski, Vermont, USA) immediately after inoculation and then at 24 h intervals over a period of 5 d until the absorbance
reached a plateau.

Heterotrophic functional diversity was assessed by calculating the mean Shannon diversity index (carbon H') of the car-
bon substrates use across the 5 d incubation period. This was done using the "diversity" function from the R package vegan
(Oksanen et al., 2022), providing an integrated measure of substrate utilization diversity over time. To evaluate patterns in
substrate use, the 31 carbon sources on the Biolog EcoPlates were categorized based on their chemical characteristics (amino
acids, amines, carbohydrates, carboxylic acids, polymers, and phenolic compounds) and macronutrient composition. Specifi-
cally, substrates were grouped as containing only C, C and N (used for calculating the nitrogen use index, NUSE), or C and
P (phosphorus use index, PUSE), following the classification by Sala et al. (2006). For each chemical or nutrient group, the
average absorbance over the 5 d incubation period was calculated and expressed as a percentage of the total AWCD, according
to the procedure described for the NUSE index in Sala et al. (2006). Biomass and AFDM were determined by filtering 10 mL
of the biofilm suspension onto pre-combusted filters (Cytiva Whatman GF/F filters, nominal pore size: 0.7 $\mu$m). Filters were
dried to quantify total dry mass and subsequently combusted for determination of AFDM. Chl a was measured according to
DIN 38409-60 (Deutsches Institut für Normung (DIN), 2019). For this, 10 mL of the biofilm suspension were filtered and Chl
a was quantified photometrically using a Cary 60 UV-Vis spectrophotometer (Agilent Technologies, Santa Clara, CA, USA).

## 2.5 Statistics

All statistical analyses were performed in R (version 4.4.0 Team, 2024). The data and R code can be found at Große et al.
(2025). Redundancy analysis (RDA) was conducted by applying the "rda" function after standardizing the data with the "de-
costand" function (both from the vegan package, Borcard et al., 2018). Parameters included in the RDA were categorized as
either biofilm characteristics (functional and structural metrics) or environmental parameters (catchment characteristics, dis-
solved macronutrient concentrations and ratios, physicochemical conditions, treatments, or sampling site). Forward selection
was then performed using the "ordiR2step" function from the vegan package in R, which incrementally adds variables to the
model based on adjusted $R^2$ and permutation tests. We specified a scope that included all variables from the global model and
constrained the selection process to ensure that only variables significantly improving model fit were retained. Significance
was assessed using 1000 permutations. Adjusted $R^2$ values were used to control for model overfitting. The optimized RDA
model included the following environmental variables: sampling site, treatment, final $NH_4$-N and $NO_X-N$ concentrations in
the mason jars, the initial DOC : DIN at the start of incubation, and the final $DOC:PO_4-P$ at the end of the experiment. In a
subsequent step, biofilm variables with loadings below 0.6 were excluded to improve model interpretability. The significance
of RDA axes and environmental variables were assessed using permutation tests via the "anova.cca" function from the vegan
package. The entire step was repeated for two RDAs performed with sub-datasets, which were split into structural (POC, PN,
PP, POC:PN, PN : PP, POC:PP, biomass, Chl a) and functional (CLPP parameters, metabolism, absolute and relative $NO_3-N$
uptake, changes in macronutrient concentrations) biofilm variables.

To investigate treatment-specific effects on absolute and relative $NO_3-N$ uptake, we tested the effects of DOC and N
addition as well as their interaction with a linear model using the "lm" function for each sampling site separately. We performed





post-hoc comparisons between treatment groups using the "emmeans" function with false discovery rate-adjusted pairwise comparisons (Benjamini and Hochberg, 1995) and set a significance threshold for the adjusted p-value of $\alpha$=0.05.

To investigate the connection between biofilm metabolism and $NO_3-N$ uptake rates, we tested relationships between GPP or CR, and absolute $NO_3-N$ uptake rates. We scaled all continuous variables ("scale" function) before testing their effects with the "lm" function and checked model assumptions with the "check_model" function from the performance package.

## 3    Results

### 3.1    Biofilm structure

Biofilm structural parameters were significantly affected by the sampling site, initial DOC : DIN and final $NH_4$-N concentration, with sampling site as main driver of differences in biofilm structure. The RDA was able to explain 73 % of the total variation and axes 1-3 were significant (axis 3 in Appendix A Fig. A2). Hatchet Creek biofilm structure was correlated with high initial DOC : DIN and was characterized by high PN : PP and POC : PP in the biofilm (Fig. 2a). The downstream site at Hogtown Creek was characterized by high biomass, POC, PN, PP, EPS, and a high POC : PN in the biofilms (Fig. 2a).

Sweetwater branch correlated with the final $NH_4$-N concentration and was characterized by high Chl a concentrations (Fig. 2a). The RDA of structure and function together is visualized in Appendix A Fig. A1.

### 3.2    Biofilm function

The biofilm functional parameters were significantly affected by the sampling sites, treatments, final $NH_4$-N concentrations, final DOC : $PO_4-P$, final $NO_3-N$ concentration, and initial rDOC : DIN, with sampling site as main driver of differences

in biofilm functional parameters. The RDA was able to explain 66 % of the biofilm functional parameters and axes 1-5 being significant (axes 3-5 in Appendix A Fig. A3). Hatchet Creek exhibited higher PUSE and NUSE indices, and was correlated with the final rDOC : $PO_4-P$ (Fig. 2b). Hogtown Creek (upstream) exhibited a higher carbon H' (Fig. 2b). Hogtown Creek (downstream) was characterized with high CR, AWCD, $U_{rel}$, and a high $\Delta$ $PO_4-P$ (Fig. 2b). Sweetwater Branch exhibited a high $U_{sed}$, GPP, $\Delta$ $NH_4$-N, and $\Delta$ $NO_X-N$ (Fig. 2b).

### 3.3    Macronutrient stoichiometric ratios

The ternary diagram visualizing the water rDOC : DIN : $PO_4-P$ revealed that the studied streams differed in their background macronutrient ratios (cn treatments in Fig. 3a). Hatchet Creek, as the most natural site with a high proportion of wetlands in the catchment, had a background rDOC : DIN : $PO_4-P$ of 126 : 2 : 1 and was located in the N depletion zone of the ternary diagram. Both Hogtown Creek sites, Hogtown Creek (upstream) with a rDOC : DIN : $PO_4-P$ of 8 : 2 : 1 and Hogtown Creek

(downstream) with a rDOC : DIN : $PO_4-P$ of 30 : 1 : 3, were located in the C and N co-depletion zone of the ternary diagram. The most urban and wastewater treatment plant influenced site Sweetwater Branch with a rDOC : DIN : $PO_4-P$ of 22 : 134 : 1 was located in the C and P co-depletion zone of the ternary diagram. The addition of DOC and N in our treatments shifted the





macronutrient ratios by reducing potential DOC and/or N limitations in Hatchet Creek and both sites at Hogtown Creek (Fig. 3a). Due to high DIN concentrations in Sweetwater Branch, DOC addition was not able to move the $rDOC : DIN : PO_4-P$

out of the DOC and P co-depletion zone and N addition did not have a large effect on the water macronutrient ratio (Fig. 3a).

To evaluate the "macronutrient access hypothesis", we visualized $U_{rel}$ (as % of the initial $NO_X-N$) depending on the initial $rDOC : DIN : PO_4-P$ and the sampling site (Fig. 3a). These data demonstrated that $U_{rel}$ was inversely related to the DIN axis (Hatchet Creek) and $PO_4-P$ axis, but positively related to the rDOC axis (Hogtown Creek). This effect occurred when comparing the four treatments of Hatchet Creek, Hogtown Creek (upstream) and Hogtown Creek (downstream), but not for

Sweetwater Branch. No consistent pattern emerged across all treatments and sites. The shift of the $rDOC : DIN : PO_4-P$ from start to end of the 24 h mason jar incubation revealed that all treatments converged at each sampling site (Fig. 3b). For Hatchet Creek, Hogtown Creek (upstream) and Hogtown Creek (downstream), the location of the $rDOC : DIN : PO_4-P$ in the ternary diagram in DOC and/or N addition treatments moved towards the one from the background treatments (cn), indicating that the added DOC and N were partially removed from the water column. The location of the $rDOC : DIN : PO_4-P$ in the ternary

diagram in all treatments of Sweetwater Branch moved from the DOC and P co-depletion zone towards the same final ratio in the DOC depletion zone. Sweetwater Branch was the only sampling site where the background treatment also exhibited changes in the macronutrient ratio.

The Redfield ratio-normalized biofilm POC : PN : PP clustered in the ternary diagram according to the sampling site (Fig. 3c). Biofilm samples from Hatchet Creek exhibited a mean POC : PN : PP of 209 : 18 : 1, were positioned near the center

of the plot, indicating a balanced elemental composition. Similarly, samples from Sweetwater Branch were clustered near the center, with a mean ratio of 66 : 7 : 1. In contrast, the POC : PN : PP from Hogtown Creek (upstream), with a mean value of 66 : 4 : 1, were in the N depletion zone of the ternary diagram. Biofilm POC : PN : PP from Hogtown Creek (downstream), characterized by a mean ratio of 196 : 11 : 1, were close to the N depletion zone.

## 3.4 $NO_3-N$ uptake rates

There was a clear response to N additions by biofilms from Hatchet Creek, as the sampling site with the highest DOC and P availability, with N additions eliciting an increase in $U_{sed}$ (t = 7.07, df = 16, p < 0.001), but there was no effect of DOC addition (t = 0.03, df = 16, p = 0.98) and no interaction of DOC and N addition (t = -0.24, df = 16, p = 0.82). There were no significant effects of N or DOC additions on $U_{sed}$ at Hogtown Creek (upstream), although there were potential trends towards an effect of DOC addition (t = 1.64, df = 16, p = 0.12) and an interaction of DOC and N addition (t = 1.51, df = 16, p = 0.15). This potential

synergistic effect is visualized as significantly increased $U_{sed}$ in the DOC and N addition treatment, compared to the other treatments (Fig. 4). Hogtown Creek (downstream) showed the same patterns in the post-hoc results (Fig. 4) with a significant positive interaction between DOC and N addition in the linear model (t = 2.70, df = 16, p = 0.016). Sweetwater Branch, the sampling site located in the DOC and P co-depletion zone and with the highest background $NO_3-N$ concentration, generally showed the highest $U_{sed}$ and no significant differences between treatments in the post-hoc results (Fig. 4). Nevertheless, there

was a significant effect of the interaction of DOC and N addition, which had a negative effect on the $U_{sed}$ at Sweetwater Branch (t = -2.80, df = 16, p = 0.013).





$U_{rel}$ was highest in treatments that did not receive any N addition at Hatchet Creek and the downstream site of Hogtown Creek (Fig. 4). For Hatchet Creek, N addition decreased the $U_{rel}$ values (t = -5.26, df = 16, p = 0.001). At Hogtown Creek (upstream) the linear model revealed a positive effect of DOC addition on $U_{rel}$ (t = 4.55, df = 16, p = 0.003, Fig. 4). $U_{rel}$

by Hogtown Creek (downstream) biofilms increased in response to the DOC addition (t = 3.02, df = 16, p = 0.008), and decreased in response to the N addition (t = -5.94, df = 16, p < 0.001), but there was no significant interaction (t = -0.95, df = 16, p = 0.36). There were no significant individual treatment effects on $U_{rel}$ for Sweetwater Branch (Fig. 4), but there was a significant negative interaction between DOC and N addition (t = -2.30, df = 16, p = 0.035).

## 3.5 Metabolism as predictor of $NO_3-N$ uptake

The linear model investigating the effect of GPP and CR on $U_{sed}$ explained the highest variance for Hogtown Creek (upstream) with 61 %, followed by Hogtown Creek (downstream) with 47 % and Hatchet Creek with 24 %, while $U_{sed}$ for Sweetwater Branch could not be explained by GPP and CR (Fig. 5). There were positive effects of GPP on $U_{sed}$ at Hatchet Creek (t = 2.14, p = 0.047, $\beta = 0.51 \pm 0.238$) and the upstream site of Hogtown Creek (t = 5.59, p < 0.001, $\beta = 0.821 \pm 0.147$). At the downstream site of Hogtown Creek, CR had a positive effect on $U_{sed}$ (t = 4.30, p < 0.001, $\beta = 0.736 \pm 0.171$). At Sweetwater

Branch $U_{sed}$ was not affected by any of the two variables.

## 4 Discussion

This study investigated the effect of short term changes in macronutrient stoichiometry on biofilm function, structure, and $NO_3-N$ uptake. We found that short term nutrient pulses favor functional biofilm reactions but did not affect structural variables, supporting our first hypothesis. Especially $NO_3-N$ uptake and metabolism as important biogeochemical processes were

positively affected by adding DOC and/or N. Furthermore, patterns of higher $NO_3-N$ uptake with high DOC : DIN and low DIN : $PO_4-P$ were visible both in the background treatments across different sampling sites and in the four treatments within a single sampling site. This partly lends support to our second hypothesis. Finally, the $NO_3-N$ uptake correlated either with GPP or with CR, which is in line with our third hypothesis.

The RDA revealed that the sampling site accounted for most of the variance in biofilm function and structure. However,

we found a significant treatment effect on the functional, but not structural parameters, supporting our first hypothesis. After 48 h of incubation, the biofilm POC : PN : PP clustered according to sampling sites. Absolute and relative $NO_3-N$ uptake ($U_{sed}$ and $U_{rel}$) showed a clear treatment effect in Hatchet Creek and in both Hogtown Creek sites. These results indicate that biofilm function responds faster to treatments than biofilm structure, a phenomenon previously described by Besemer (2015). According to Proia et al. (2012) and Liu et al. (2015), treatment-specific differences were observed in bacterial abundance

and biofilm function, whereas algal abundance did not show such differences. The different reaction times between structural and functional variables underscore the importance of distinguishing between function and structure. For instance, periphyton biomass did not linearly correlate with N assimilation or NPP, which may lead to wrong interpolations (Schlenker et al., 2024).



This short-term functional adaptation supports the idea that less impacted streams can buffer nutrient pulses to some extent through functional adjustments.

## 4.1 Short-term changes of DOC : N : P imbalance as predictors for $NO_3-N$ uptake

We hypothesized the highest $U_{rel}$ for biofilms incubated in treatments of high DOC : DIN and low DIN : $PO_4-P$ in the water column. The rDOC : DIN : $PO_4-P$ fulfilling these requirements were located in the N-depletion zone of the ternary diagram (Fig. 3). The highest $U_{rel}$ for the background treatments of the four sampling sites (cn) was measured for Hatchet Creek, which had a background rDOC : DIN : $PO_4-P$ located in the N-depletion zone of the ternary diagram. The second highest $U_{rel}$ were measured for both sites at Hogtown Creek, which had a background rDOC : DIN : $PO_4-P$ located in the C and N co-depletion zone of the ternary diagram. The lowest $U_{rel}$ was measured for samples from Sweetwater Branch, with a background rDOC : DIN : $PO_4-P$ located in the C and P co-depletion zone of the ternary diagram.

For Hatchet Creek and both sites at Hogtown Creek, a consistent pattern was observed when comparing the four treatments within each site: treatments with higher rDOC : DIN and lower DIN : $PO_4-P$ showed higher $U_{rel}$ values than treatments with lower rDOC : DIN and higher DIN : $PO_4-P$. In other words, $U_{rel}$ increased as the rDOC : DIN : $PO_4-P$ of a treatment approached the N-depletion zone on the ternary diagram. This pattern, however, did not hold for Sweetwater Branch. Due to its high background DIN concentrations, the treatments at this site caused only minor shifts in the rDOC : DIN : $PO_4-P$, keeping all treatment points clustered in the DOC and P co-depletion zone. Consequently, there was no clear relationship between the macronutrient ratio and $U_{rel}$ at Sweetwater Branch.

The biofilms at Sweetwater Branch growing under high chronic DIN loadings generally had the highest absolute $U_{sed}$, the lowest $U_{rel}$, and no treatment effects (Fig. 4). Here, the high chronic DIN loading probably resulted in long-term adaptation of uptake. Biofilms in more eutrophic systems seem to be able to deal better with chronic loadings than with short term pulses (Weigelhofer et al., 2018), potentially leading to a slower adaptation of the biofilms from Sweetwater Branch to changes in macronutrient ratios and concentrations. Such high absolute $NO_3-N$ uptake ($U_{sed}$) with low uptake efficiency ($U_{rel}$) implies N saturation of biofilm nutrient uptake, and has already been described as an increased $NO_3-N$ uptake length and therefore a decreased uptake efficiency in streams of higher $NO_3-N$ concentration (Mulholland et al., 2009). It is important to consider that Sweetwater Branch was the only site with excess N, where all rDOC : DIN : $PO_4-P$ were located in the DOC and P co-limitation zone. Prior research has revealed that substantial increases in heterotrophic $NO_3-N$ uptake can be achieved if the stoichiometry is shifted outside of the DOC and P co-limitation zone (Große et al., 2025b). When insufficiently changing the rDOC : DIN : $PO_4-P$ stoichiometry to move a stream outside of the zone of DOC and P co-depletion, other drivers – such as light, hydromorphology, and fine sediment clogging – may be more important for $NO_3-N$ dynamics (e.g. Sunjidmaa et al., 2022, 2025).

According to the "macronutrient access hypothesis" we would have expected a clear gradient with the highest $U_{rel}$ in the upper part of the N-depletion zone in the ternary diagram. Our data showed the highest $U_{rel}$ value in the C addition treatment of Hogtown Creek (downstream), but following the location of the rDOC : DIN : $PO_4-P$ in the ternary diagram, the "macronutrient access hypothesis" would predict a higher uptake in the cn treatment of Hatchet Creek. This mismatch could



be caused by several factors: First, the "macronutrient access hypothesis" was originally developed for heterotrophic microbes in planktonic microcosms with stream water bacteria (Graeber et al., 2021). We here investigated $U_{rel}$ in more complex and structured biofilm communities, which contain both autotrophs and heterotrophs (Battin et al., 2016). The positive correlation

of GPP and $U_{sed}$ for Hatchet Creek and Hogtown Creek (upstream) suggests that $U_{sed}$ at these sites is driven by autotrophs, which makes rDOC a less important factor for $NO_3-N$ uptake. This is further supported by the fact that the addition of DOC had no effect on $NO_3-N$ uptake at Hatchet Creek. As Hogtown Creek (downstream) was the most heterotrophic sampling site (Fig. 2B) and $U_{sed}$ correlated with CR (Fig. 5), we had expected the strongest treatment-effect of DOC and N addition, which is in line with our results.

Second, the incubation time of 48 h may have been too short to change biofilm structure. This is supported by the biofilm POC : PN : PP, which were primarily grouped according to the sampling sites, with the treatments being less important (Fig. 3c). Compared to a planktonic community, biofilms have a complex, three-dimensional structure. As a consequence, more mature, thicker biofilms need longer incubation times to make sure that macronutrients are transported into deeper layers, especially in low-flow environments (Battin et al., 2003; Weigelhofer et al., 2018). Accordingly, we can conclude that short-

term pulses of 48 h incubation time were enough to see treatment-specific differences in $NO_3-N$ uptake for each sampling site but not long enough for the treatments to offset site-specific effects.

A third potential explanation is that we did not measure rDOC concentrations directly. Instead, we estimated the background bioavailability of DOC by combining catchment-specific land use data with DOC bioavailability data from Stutter et al. (2018). Direct measurement after an incubation would be more precise and potentially alter the location of the rDOC : DIN : $PO_4-P$

in the ternary diagram. We tested the sensitivity of the rDOC : DIN : $PO_4-P$ positions in the ternary diagram (see Appendix B Fig. B1 and B2), visualizing the positions corresponding to the highest and lowest realistic bioavailability values reported by Stutter et al. (2018). This range shows that the position does not change much for most of the sites and treatments, but the cn and cN treatments at Hatchet Creek exhibited greater variability in their position in the ternary diagram. Assuming the lowest DOC bioavailability of 2.4 %, as found in moorland waters, this would explain the observed mismatch of lower uptake in the

cn treatment at Hatchet Creek than predicted by the "macronutrient access hypothesis". On the other hand, the fact that there was no DOC effect on $NO_3-N$ uptake at Hatchet Creek suggests that there was no DOC limitation of heterotrophic $NO_3-N$ uptake in the cn and Cn treatments (Fig. 4).

We used acetate as a non-natural DOC source due to its ease of handling and comparability with other studies (Bechtold et al., 2012; Oviedo-Vargas et al., 2013). DOC concentrations in C addition treatments declined strongly during incubation,

likely due to respiratory removal. Future studies might use more natural DOC sources such as leaf or soil leachates, or algal exudates (Duan et al., 2014; Hansen et al., 2018; Lyon and Ziegler, 2009). Using mason jars for incubation alters hydrological conditions, creating lentic environments with longer retention times, closer to intermittent streams, compared to natural stream flow. Future research should also examine the fate of assimilated N. We measured only $NO_3-N$ uptake, but N may be released in inorganic forms, removed via denitrification, or converted to recalcitrant DON (O'Brien et al., 2012; Stutter et al., 2018; von

Schiller et al., 2009).




## 4.2 Metabolism

Positive relationships of $U_{sed}$ with GPP and CR have been observed previously (Beaulieu et al., 2014; Hall Jr. et al., 2009; Hall and Tank, 2003; Mulholland et al., 2008). The streams in our study react differently depending on site-specific biofilm characteristics or anthropogenic catchment influences. We initially hypothesized that increasing GPP and CR leads to increased
$U_{sed}$. Indeed, $U_{sed}$ correlated positively with GPP at Hogtown Creek (upstream) and Hatchet Creek, and with CR at Hogtown Creek (downstream). At Sweetwater Branch, however, no correlation of $U_{sed}$ with GPP or CR was found.

The positive relationship between GPP and $U_{sed}$ at Hatchet Creek and Hogtown Creek (upstream) suggests a strong autotrophic contribution to $U_{sed}$, or stimulation of heterotrophs by photosynthesis products (Heffernan and Cohen, 2010; Romani and Sabater, 1999). For Hogtown Creek (downstream), many measured GPP values were negative, which is not biologically
possible. The exceedingly low (below detection) GPP from Hogtown Creek (downstream) reflects the slow-moving flow and organic-rich sediments present at this site, leading to highly heterotrophic metabolic activity reflected by the RDA and at the reach-scale (Taylor, 2023). Therefore, we set these values to 0, which explains why we were unable to demonstrate a relationship between GPP and $U_{sed}$ for Hogtown Creek (downtown) biofilms. To avoid light limitation of photo-autotrophs, all treatments were exposed to the same light intensity. The mean PAR intensities in Table 1 shows that the natural conditions can
be light limited by natural shading of vegetation. Considering that Wymore et al. (2016) found PAR to be more influential on $NO_3-N$ uptake than GPP, future studies should validate autotrophic effects under natural PAR conditions. Storm-driven DOC and N pulses can also increase turbidity (Steward et al., 2006), potentially reducing light availability for autotrophic $NO_3-N$ uptake.

Hogtown Creek (downstream) is the most heterotrophic site, indicated by a correlation with CR in the RDA. We found a
positive correlation between CR and $U_{sed}$. This site also exhibited the highest levels of structural variables (biomass, POC, PN, PP, AFDM, EPS). The Hogtown Creek (downstream) sampling site is located within a forest that drains urban areas, resulting in a large quantity of particulate organic matter. This could have provided a natural surface for the colonization of biofilms, as well as an additional C source for heterotrophs (Dodds and Smith, 2016; Johnson et al., 2009). Combined with significantly increased $U_{sed}$ in the treatments of DOC and N addition (Fig. 4), higher AWCD and higher heterotrophic functional diversity
(carbon H', Fig. 2b), these results suggest a strong heterotrophic contribution. At Sweetwater Branch, the most urban site with excess N availability, we measured the highest GPP rates. We did not find any effects of DOC or N addition on $U_{sed}$, possibly due to the P limitation or the high $NH_4-N$ concentration, which is preferred over $NO_3-N$ (Cejudo et al., 2020). We can conclude that sampling site specific biofilm structure helps to explain the effect of GPP and CR on $NO_3-N$, supporting the third hypothesis.

## 5 Conclusions

Our study revealed that biofilms are capable of rapidly responding to short term pulses by increasing biofilm $NO_3-N$ uptake, providing a buffering mechanism that could reduce downstream nutrient transport. The different effect intensity between biofilm structure and function shows that functions react faster when water macronutrient imbalances change and that these



changes can be predicted by considering the water rDOC : DIN : $PO_4-P$. Furthermore, high anthropogenic impact through

wastewater decreases streams' capacity to deal with high nutrient pulses. Predictions of self-cleaning potential and buffer capacity of short term nutrient pulses should thus always consider the streams' anthropogenic loading background. Connecting metabolism and N cycling in biofilms enabled us to more precisely predict ecological consequences for specific stream reaches. Since changes in land use affect macronutrient ratios in streams (Stutter et al., 2018; Wachholz et al., 2023), and especially the reconnection of wetlands to streams can increase DOC concentrations and enhance denitrification (Hansen et al., 2018, 2016),

these processes may influence the connection between $U_{sed}$ and stream metabolism. As a result, more heterotrophic streams are expected to show a stronger increase in $NO_3-N$ uptake in response to land use changes compared to less heterotrophic streams. In contrast, more autotrophic systems will show decreased $NO_3-N$ uptake when shading is increased due to increased canopy cover, burial or other constructions (Pennino et al., 2014; Reisinger et al., 2019). Additionally, the contribution of autochthonous DOC to increase heterotrophic $NO_3-N$ uptake can be influenced by light intensity and should therefore be

considered when predicting changes in biogeochemical cycles with land use changes. Taking into account future changes in land use, such as the restoration of wetlands and the increase in canopy cover in urban ecosystems (European Commission. Directorate General for Environment., 2025), it can be suggested that an intensive monitoring of metabolism and N cycling in headwaters during succession can help us to understand the connections between the biogeochemical cycles described in our study.

*Code and data availability.* The data and code are available online at https://doi.org/10.5281/zenodo.15878295

## Appendix A: RDA for structural and functional parameters

The RDA with selected environmental factors and the functional and structural biofilm parameter of loading > 0.6 resulted in an adjusted $R^2$ of 0.7. Sampling site explained most of the variance, followed by treatment, initial start DOC : DIN and mason jar $NH_4$-N as significant environmental parameters. Furthermore, axes 1-4 were significant, explaining 42 %, 17 %, 10 % and

2 % of the total variance (Fig. A1).

The plots of the RDA visualize that the sampling site explains most of the variance within biofilm parameters. Axes 1 and 2 were able to separate the samples by sampling site. Samples from Hatchet Creek correlate with PUSE and biofilm N : P. Hogtown Creek (upstream) correlates mainly with a carbon H', while Hogtown Creek (downstream) correlate with biofilm C : N, the mason jar $PO_4-P$ difference, POC, AFDM, EPS, PN, CR, biomass and AWCD. Sweetwater Branch as the most

impacted site correlates with $U_{sed}$ and $U_{rel}$, GPP, Chl a as biofilm parameters, and difference of $NO_X-N$ in mason jars, as well as mason jar $NH_4$-N concentrations as environmental parameter. RDA axes 3 and 4 were able to divide the samples according to N addition, with the treatments without N addition on one part of the RDA axis 4 and the treatments with N addition on the other part.



**Appendix B: Sensitivity analysis for rDOC**

We were not able to measure the bioavailable fraction of the DOC in the background samples of the four streams. Consequently, we have estimated this fraction by using literature data. The DOC concentrations of the background samples (BG) and the treatments (cn, cN, Cn, CN) are in the same order of magnitude (Fig. B1). Since these are the bulk and not the bioavailable fractions, missing large bioavailable parts in this method would result in a steep decrease of DOC concentrations during the incubation, as we have seen for the added acetate. This was not the case and visualized that we did not miss a larger amount of

bioavailable DOC by applying the land use specific bioavailability factors according to the catchment characteristics.

We calculated the minimum and maximum DOC availability values according to the literature data. We used 2.4 % for the lowest value for moorland source water, and 44.8 % for the highest value for Sweetwater Branch (sewage effluent). For both sites at Hogtown Creek, we used 17.1 % for urban runoff, and for forest soil water at Hogtown Creek, we assumed 22.4 % (Stutter et al., 2018). Analysis of the positions in the ternary diagram shows that the $DOC : DIN : PO_4-P$ did not vary in the

depletion zone for the lowest and highest bioavailability at Sweetwater Branch and both sites at Hogtown Creek (Fig. B2). The treatments without C addition at Hatchet Creek (cn and cN) showed the largest variability in the ternary diagram. For the cn treatment, the position of the $DOC : DIN : PO_4-P$ are both still in the N-depletion zone, but on different ends. The position in the ternary diagram of the cN treatment of Hatchet Creek can be in the C-depletion zone (lowest bioavailability), but with increasing bioavailability, the position can be in no depletion zone or even in the P-depletion zone (highest bioavailability).

*Author contributions.* Conceptualization: all authors. Data curation: AG. Formal analysis: AG. Funding acquisition: DG, DB and AJR. Investigation: AG with support from AJR. Methodology: AG with support from NP and AJR. Project administration: AG. Resources: AJR, NP, and DB. Supervision: DG and AJR. Validation: AG, AJR, and DG. Visualization: AG, AJR, and DG. Writing - original draft: AG. Writing - review and editing: all authors.

*Competing interests.* The contact author has declared that none of the authors has any competing interests.

*Acknowledgements.* This work was done within the Helmholtz International Research School TRACER (HIRS-0017) funded by the Initiative and Networking Fund of the President of the Helmholtz Association. We thank U. Schlägel for the statistical support, the UFZ water analytics laboratory (GEWANA), in particular A. Hoff and I. Siebert, but also A. Ballmann and I. Locker for their excellent technical assistance. Furthermore, we thank the members of the Reisinger lab, in particular C. Harris and E. Taylor for their assistance with field work, A. Goeckner and J. Reimer for guidance with lab analysis. We acknowledge the assistance of DeepL Write in improving the language of the

first draft of this manuscript.



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



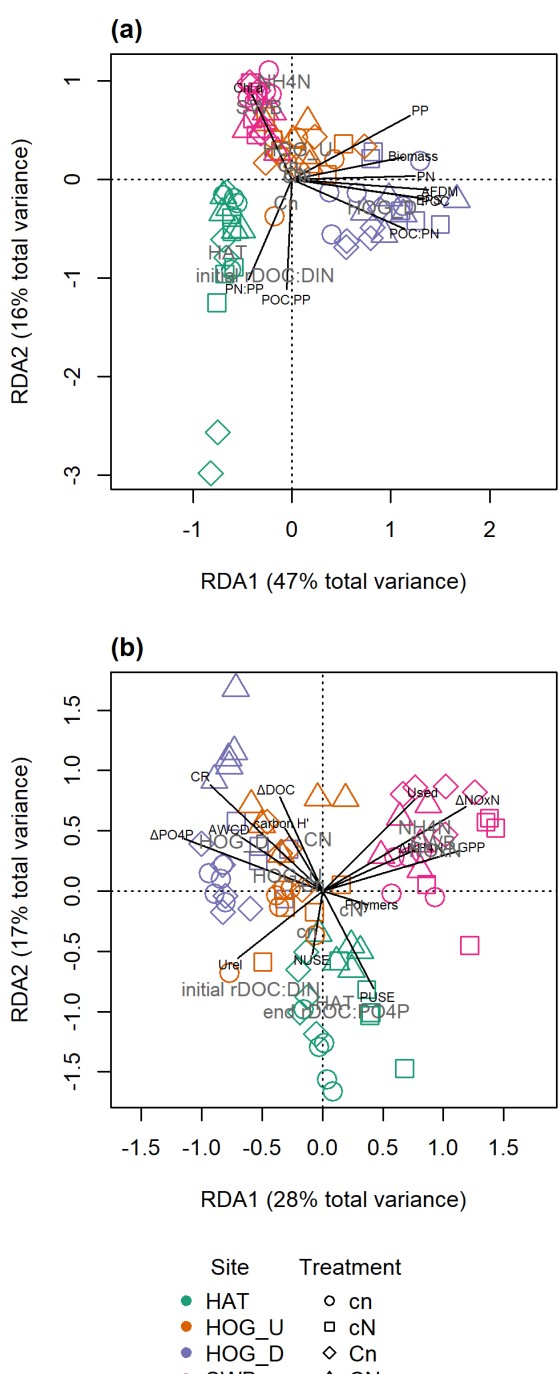

**Figure 2.** RDA of the biofilms structure (a) and function (b). Environmental factors are in grey, biofilm functional parameters are in black. The points represent the mason jars with sampling sites in different colors and the shape represents the treatment. HAT = Hatchet Creek, HOG_U = Hogtown Creek (upstream), HOG_D = Hogtown Creek (downstream), SWB = Sweetwater Branch.



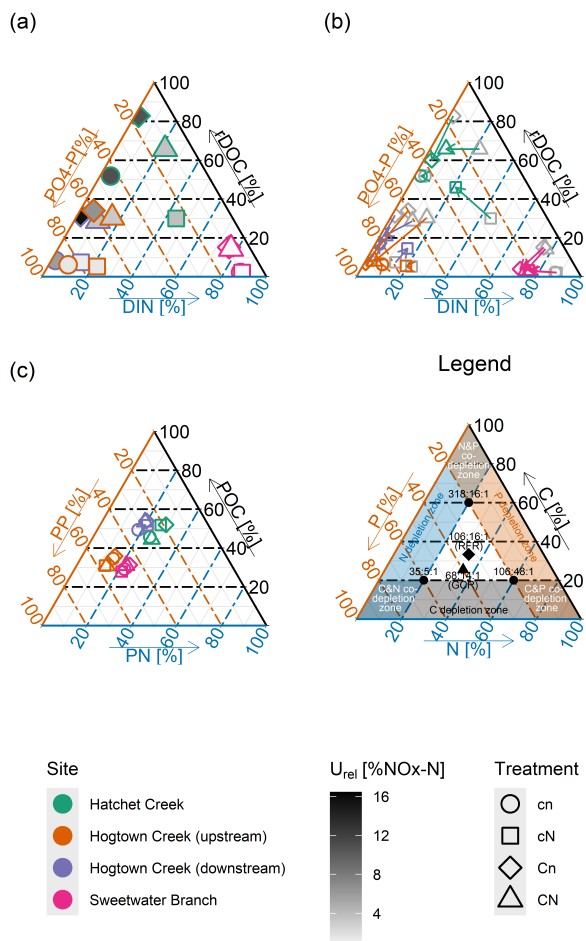

**Figure 3.** Ternary diagrams of initial (a) and final mason jar water macronutrient ratios (b) and biofilm ratios (c) with sampling site (color) and treatment information (shape). Panel a shows initial ratios with filling color-coded for relative nitrate-N uptake ($U_{rel}$). Panel b shows the shift of water macronutrient ratios from initial (grey) to end of the incubation. Panel c shows the ternary diagram of the biofilm C : N : P. All points are mean values per treatment (n = 5).





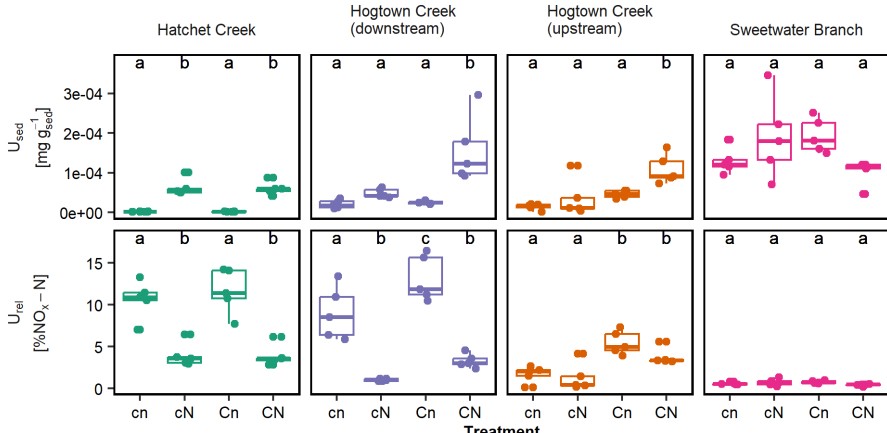

**Figure 4.** Absolute $NO_3-N$ uptake rate ($U_{sed}$) and relative $NO_3-N$ uptake ($U_{rel}$) depending on sampling site and treatment.

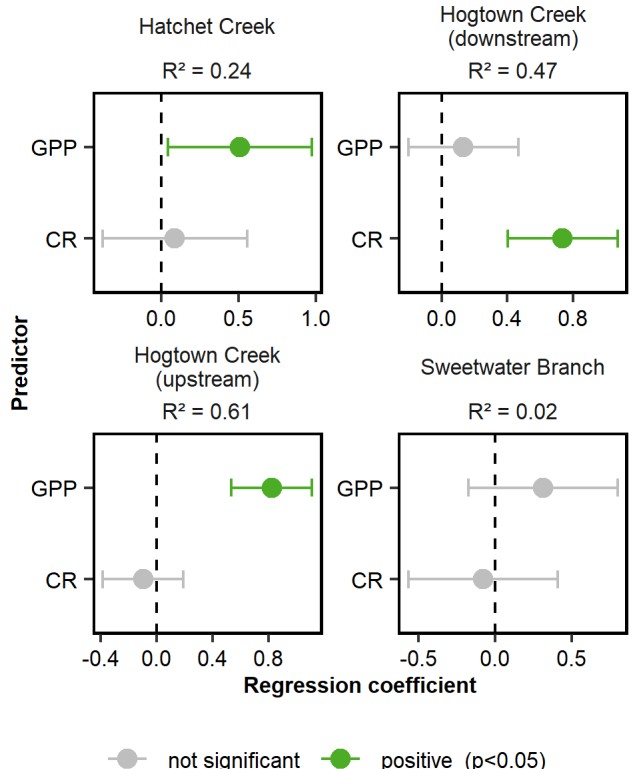

**Figure 5.** Results of linear regression model for each sampling site investigating the effect of GPP and CR on $U_{sed}$.





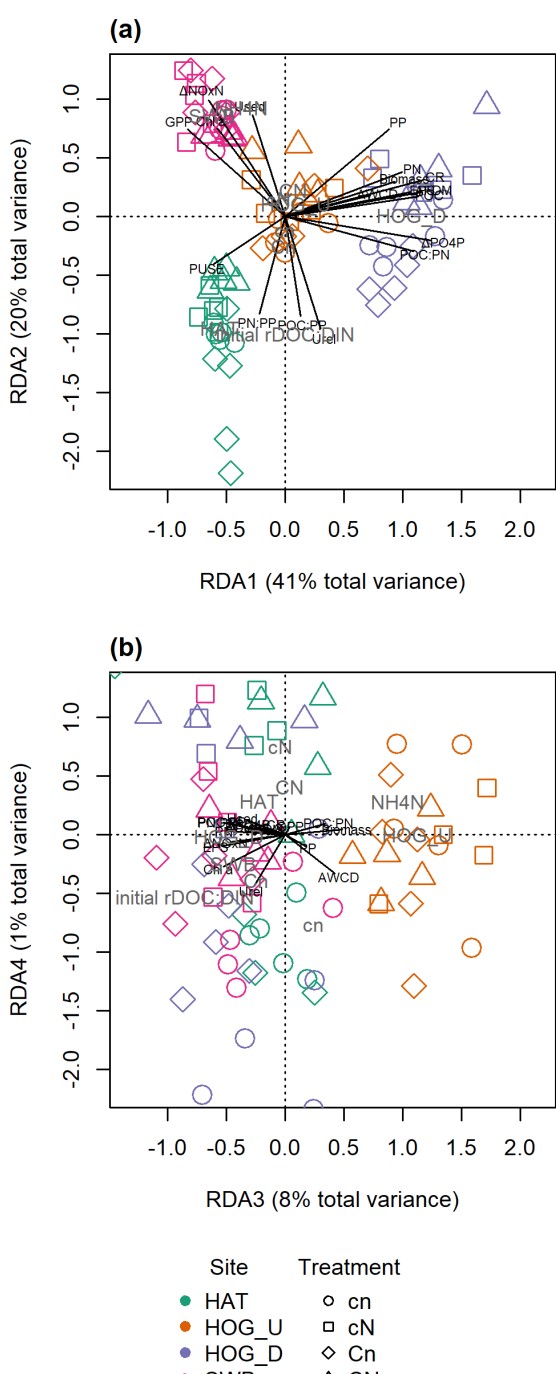

**Figure A1.** RDA of the biofilm structure and function for axes 1 and 2 (a), and axes 3 and 4 (b). Environmental factors are in grey, biofilm functional parameters are in black. The points represent the mason jars with sampling sites in different colors and the shape represents the treatment. HAT = Hatchet Creek, HOG_U = Hogtown Creek (upstream), HOG_D = Hogtown Creek (downstream), SWB = Sweetwater Branch.




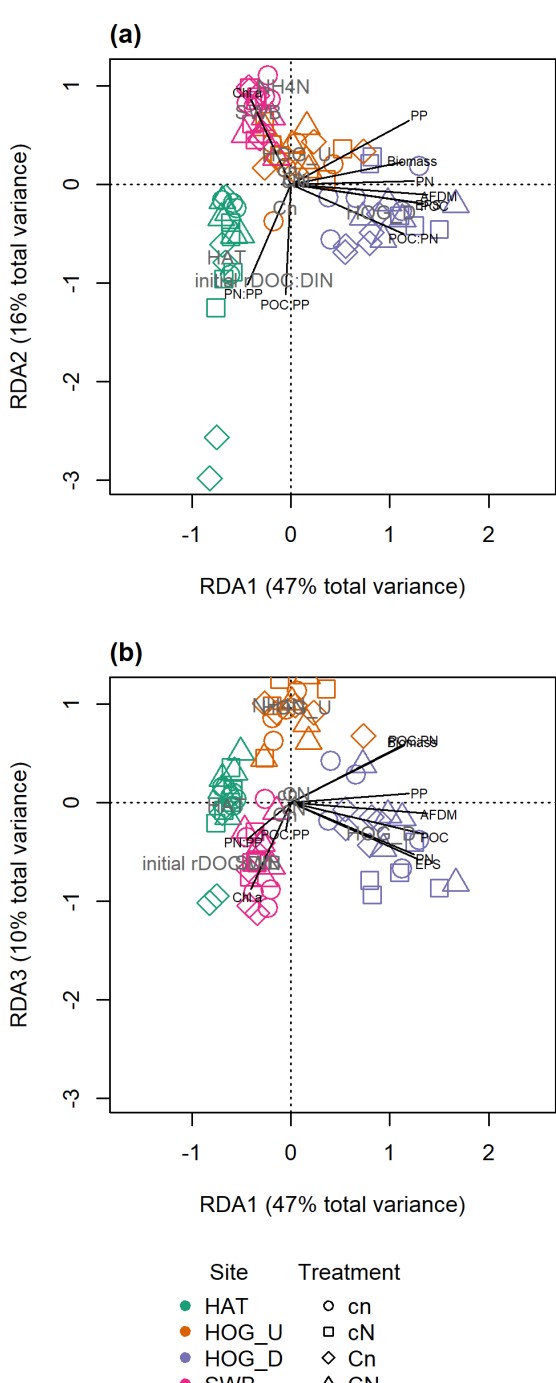

**Figure A2.** RDA of the biofilm structure for axes 1 and 2 (a), and axes 1 and 3 (b). Environmental factors are in grey, biofilm functional parameters are in black. The points represent the mason jars with sampling sites in different colors and the shape represents the treatment. HAT = Hatchet Creek, HOG_U = Hogtown Creek (upstream), HOG_D = Hogtown Creek (downstream), SWB = Sweetwater Branch.



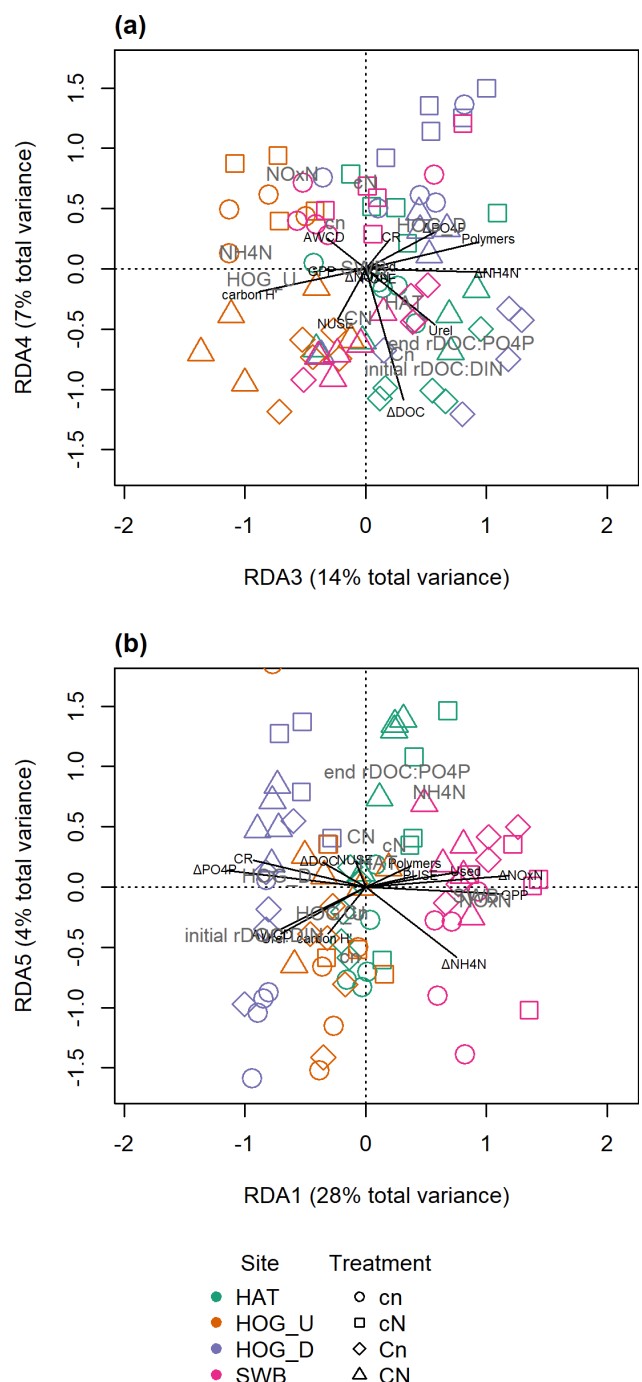

**Figure A3.** RDA of the biofilm function for axes 3 and 4 (a), and axes 1 and 5 (b). Environmental factors are in grey, biofilm functional parameters are in black. The points represent the mason jars with sampling sites in different colors and the shape represents the treatment. HAT = Hatchet Creek, HOG_U = Hogtown Creek (upstream), HOG_D = Hogtown Creek (downstream), SWB = Sweetwater Branch.



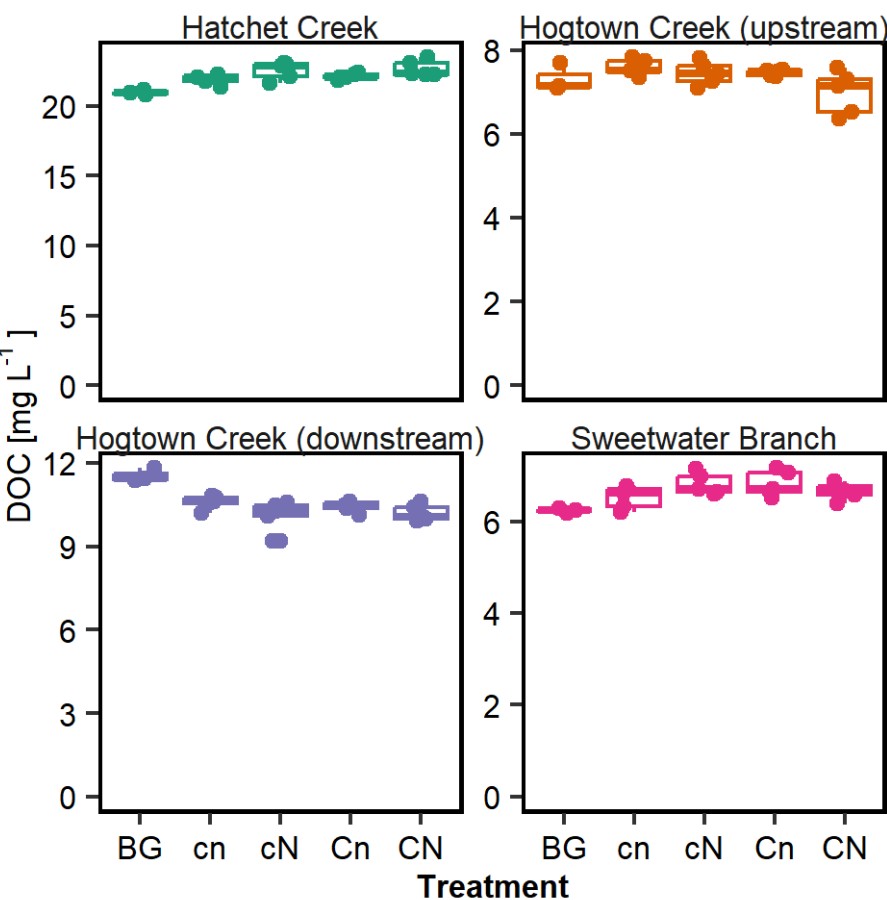

**Figure B1.** Boxplot of DOC concentrations for each sampling site. BG = initial background concentration, and final treatment concentrations in mason jars: cn = no addition, cN = N addition, Cn = DOC addition, CN DOC and N addition. Note that y-axes have different scaling.





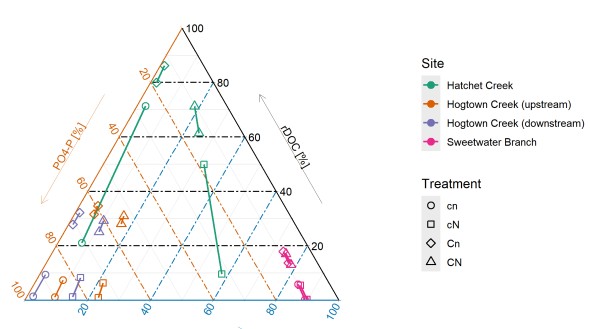

**Figure B2.** Sensitivity analysis for the position of the Redfield Ratio normalized DOC : DIN : $PO_4-P$ in the ternary diagram. Initial treatment concentrations in mason jars: cn = no addition, cN = N addition, Cn = DOC addition, CN DOC and N addition