# Peer review of "Acute changes in macronutrient stoichiometry alter nitrate uptake in benthic biofilms"

_EGUsphere, 2025_

## Author Comment (AC1)

**General response**

Dear Reviewers and Editor,

Thank you for recognizing the potential of our manuscript and for providing constructive feedback to help improve its quality. We appreciate your detailed comments and agree with most of the points raised. We are committed to revising the manuscript thoroughly to enhance clarity and strengthen the overall message. Please find our detailed responses to each comment below.

**Reviewer 1:**

**Comment 1:**
The paper investigates benthic biofilms as hotspots for macronutrient cycling in Florida headwater streams. It presents an incubation experiment to assess biofilm nitrate uptake under varying macronutrient ratios. The study has the potential to make a valuable contribution to the literature; however, it contains several major flaws that need to be thoroughly addressed. The manuscript can be reconsidered for publication only after these primary issues are properly resolved.

**Response to comment 1:**
Thank you for recognizing the potential of our study. We agree that substantial improvements are needed, and we are committed to addressing all major concerns to strengthen the manuscript.

**Comment 2:**
Introduction: The motivation for the study is not clearly presented. It is unclear why these specific sites were selected and what unique characteristics make them suitable for this type of investigation. The introduction does not sufficiently explain how this work contributes to the broader literature, how findings from Florida watersheds can be extrapolated to other stream systems globally, or what the key biogeochemical implications of the study are. These elements are missing despite the introduction being very lengthy. Although hypotheses are stated, they are not adequately introduced or explained.

**Response to comment 2:**
Thank you for this comment. We agree that the introduction needs to better articulate the study's motivation, its contribution to the broader literature, and its biogeochemical implications. In the revision, we will:

- Clearly explain the rationale for site selection, emphasizing their high natural phosphorus loadings and rare nitrogen-limited conditions.
- Refine the presentation of our hypotheses to ensure they are clearly introduced and logically connected to the study objectives.
- Improve the discussion on how findings from Florida streams can inform understanding of nutrient dynamics in other streams.

**Comment 3:**
Experimental: A map of the study sites is required. Several sites are described as being more anthropogenically impacted than others, but this distinction is not clearly illustrated. A map or other visual representation should

be included to show the relative locations of the sites, along with the corresponding land-use characteristics. This would allow readers to understand the spatial context of the study and how site differences may influence the observed patterns.

**Response to comment 3:**
We agree and will include a map showing the spatial distribution of the study sites with their land-use characteristics in the catchment. This will complement the land-use data presented in Table 1 and provide readers with a clearer spatial context.

**Comment 4:**
Results: None of the measured parameters are clearly presented in the manuscript. I could not find results for any variables measured before and after the incubations. While the dataset is available in an online repository, the manuscript does not reference or describe any of these measurements. This makes it impossible to follow what was done during the incubation experiments or how the parameters changed over time. Results section heavily refers statistical tests, without presenting the underlying data, which undermines the clarity and interpretability of the study.

**Response to comment 4:**
We agree that the results section can be improved. To address this, we will:

– Include a table summarizing all measured parameters for each sampling site and treatment.
– Add pre-incubation measurements to Table 2 to provide a clear picture of initial stream conditions. This will allow readers to better understand the experimental setup and interpret the statistical analyses we present.

**Comment 5:**
Discussion: I stopped reading the paper after noting that the empirical results were not actually presented in the results section. Once the results section is corrected and the measured data (before and after incubations) are fully reported and displayed, the discussion must be thoroughly rewritten. Such that: it should be grounded in the actual measurements including means, ranges, changes etc. rather than starting from statistical interprations. Every interpretive statement must point to the corresponding table, figure, or data. Statistical models should be presented and used as complementary to support the observed patterns.

**Response to comment 5:**
As mentioned earlier, we will include a table showing the mean and standard deviation for each parameter. Since there are 28 biofilm parameters per mason jar and 43 environmental parameters per sampling site, we decided to use an advanced multivariate statistical approach, such as RDA, to focus only on the factors that are important for the key message of this manuscript. We agree that the results and discussion should be based on actual measurements because the statistics presented correspond to data from actual measurements. We hope that including an additional table will provide sufficient information. Furthermore, we will connect every statement to a table or figure or provide the results of a statistical test.

**Reviewer 2:**

**Comment 6:**

In this study, Große et al performed a lab experiment to monitor how biofilm structure and function affect nutrients uptake. The amount of work for getting the results and the number of parameters measured is remarkable. However, the authors did not manage to aggregate those results in a coherent way. I think the manuscript has an immense potential, but the authors also need to put a lot of effort to improve the manuscript. On a separate note, I would like to apologise for not providing a timely review.

**Response to comment 6:**

Thank you for acknowledging the effort involved in this study and for recognizing its potential. We agree that the manuscript requires substantial improvement to present the results in a more coherent and accessible way. Your comments are very helpful in guiding these revisions.

**Comment 7:**

The manuscript would be far easier to understand and the results far easier to appreciate if the authors invest more time in the writing. The manuscript is understandable, but the flow of ideas and structure of sentences often sometimes unclear. Sentences containing random facts are placed within the text and it is unclear why. The text is wordy in general. I am pretty sure the authors can significantly improve the quality of the writing.

**Response to comment 7:**

Thank you for this comment. We agree that the manuscript needs clearer structure and improved flow. In the revision, we will invest more time to streamline the text, remove unnecessary details, and ensure that each section builds logically toward the main message.

**Comment 8:**

The number of replicates is not mentioned in the methods section. I assume it is 5 based on the results section but it is important to clarify that in the methods section. It would be great to have the full factorial design better explain in Figure 1. The authors also need to provide further information about why they used different methods at the "startpoint" and "endpoint", especially for Nox and NH4 which are central for the study. It is unclear how gastight the mason jars were. Have the authors tested that? Also, it is hard to believe such a jar was completely free of air bubbles. I assume the authors did not put the jars upside down to check for the presence of air bubble, because that would severely compromise the experiment. Therefore, it is unclear how the authors can guarantee there was no air bubbles, and more importantly that the same amount of air bubble was present between the treatments. Typically, we use screw cap with cone insert/liner to ensure no air bubbles are present. This could explain why the authors find that "many measured GPP values were negative, which is not biologically possible" besides the typical issue with instrument sensitivity.

**Response to comment 8:**

Thank you for your comment. We will include the number of replicates in the Methods section and improve Figure 1 to better explain the treatment design.

We apologize for the confusion regarding the water chemistry methodology. The same methods and equipment were used for the start (site conditions) and end of incubation $NH_4$, $NO_x$, and $PO_4$ samples, but the analyses were conducted in different laboratories. The initial samples were analysed by a contract lab with TN and TP

capabilities. Ultimately, we were able to access a TOC-L TOC and TN analyser. Therefore, we no longer needed the TN data from the contract lab, but we no longer had the samples to run on the same analyser. However, both analyses used the same methods, and both were conducted on a Seal discrete analyser. We can simplify this by summarizing both paragraphs in the revised manuscript.

Regarding the mason jar setup, we have previously used this experimental approach with mason jars (Gallagher and Reisinger, 2020 https://doi.org/10.1016/j.scitotenv.2019.135728 ; Reisinger et al., 2021 https://doi.org/10.1002/ecs2.3527 ) with and without inversion. We found that mason jars sealed in this way prevent air bubbles in the incubation chamber as well as centrifuge tubes, which we also use extensively. Furthermore, all the negative GPP values were from one specific site, suggesting that something was unique to that site rather than a systematic methodological issue that would be expected to affect all sites evenly. Since the sediment was sandy and mobile under natural conditions, we could invert the mason jars without disturbing the sediment structure because no clear biofilm mat had developed. Consequently, we were able to prove the absence of air bubbles.

**Comment 9:**
In the results section, the authors need to report the results of the test when saying something was significantly affected. For instance, section 3.1 and 3.2 start by saying there are significant results, but the p-value, test statistics, and degree of freedom are not presented. Oddly enough, such information is present in section 3.4 and 3.5. Also, where are the results for the Ecoplate and the many parameters measured by the authors? It feels like a lot of information is missing.

**Response to comment 9:**
Thank you for pointing this out. We will include the missing test statistics in sections 3.1 and 3.2. Regarding Ecoplate results, these were not significantly different among treatments, so we only included them in the RDA. However, based on your suggestion and Reviewer 1's feedback, we will add a table summarizing these results for completeness.

**Comment 10:**
The discussion is repeating the results section quite a lot. The discussion does not manage to bring all results together and does not help the reader understanding how this study advances our understanding of nitrate uptake in benthic biofilms. The discussion describes why some of the results might or might not be accurate but rarely brings us back to what is happening in nature.

**Response to comment 10:**
Thank you for this helpful feedback. We will revise the discussion to avoid repetition and focus on synthesizing the findings. The revised discussion will emphasize how the results advance understanding of nitrate uptake in benthic biofilms and their ecological relevance, linking experimental outcomes to natural stream processes.

Further comments:

**Comment 11:**
In the abstract, it is really hard to follow the logic between the first and second sentence. The abstract is not really down to the point. The study is about nitrate cycling in streams exposed to different anthropogenic pressure. The abstract does not help us understand that until we reach the results sentences. Please work on this and try to refocus the introduction sentences within the abstract.

**Response to comment 11:**
Thank you for this comment. We agree that the abstract needs to clearly state the focus on nitrate cycling under varying anthropogenic pressures. We will rewrite the opening sentences to emphasize this and improve the logical flow between background, objectives, and results.

**Comment 12:**
Line 19: It is unclear why this idea is brought up. It feels disconnected from the previous and following sentence.

**Response to comment 12:**
Thank you for this comment, we fully agree and will remove this sentence as it does not help to build up the storyline.

**Comment 13:**
Line 24: the term extracellular polymeric substances was coined much earlier than Battin et al., 2016. Instead of having a reference to Battin et al 2016 after EPS, maybe combine the reference to Battin et al 2016 and Freeman and Lock at the end of the sentence. Otherwise, without necessarily citing the original 1982 paper, consider citing some of the older literature besides Battin et al 2016 to avoid giving a false impression that EPS are something relatively new when they are not.

**Response to comment 13:**
Thank you for this comment, we fully agree and will follow your suggestion.

**Comment 14:**
Line 25: The sentence "The most reactive zone of stream sediments are the upper 2 cm, also known as the benthic biolayer (Knapp et al., 2017)." Is disconnected from the previous and following sentence.

**Response to comment 14:**
Thank you for this comment, we fully agree and will remove this sentence as it does not help to build up the storyline.

**Comment 15:**
Line 47: The Redfield ratio is related to marine plankton so it is unclear why we would expect to get such ratio in headwater streams. Maybe I missed something.

**Response to comment 15:**
Thank you for raising this point. While the Redfield ratio originates from marine plankton studies, similar stoichiometric patterns have been observed in benthic communities (e.g., Hillebrand & Sommer, 1999 https://doi.org/10.4319/lo.1999.44.2.0440). We will clarify this in the text and explain why the Redfield ratio can be used as a reference for optimal nutrient conditions for biofilms as well.

**Comment 16:**
Line 107: Maybe add "specifically" or similar at the beginning of the second sentence. Initially, I thought there was no information about how the calculations were done.

**Response to comment 16:**
Thank you for this suggestion. We will add "for this purpose" to improve clarity and connection between sentences.

**Comment 17:**
Line 244: It is unclear what is meant by Biofilm structure. This was not presented in the methods. The methods (preferred) or the results section need to clearly indicate which parameters are used for characterising. We cannot read anything in Fig. A1.

**Response to comment 17:**
Thank you for this comment, we described this in the statistics section of the methods. To link this clearer, we will add this reference in the figure captions. We will also improve the visibility of the labels on the RDAs.

**Comment 18:**
Line 393: "We used acetate as a non-natural DOC source" Acetate is a natural DOC source, and its degradation leads to important methane flux. It is unclear why the authors claim acetate is non-natural.

**Response to comment 18:**
Thank you for pointing this out. You are correct that acetate occurs naturally and contributes to biochemical processes. We did not intend to imply that acetate itself is unnatural, but rather that using pure acetate at the concentrations applied in our experiment does not reflect the composition of natural DOC pulses. Natural DOC inputs are typically complex mixtures in which acetate is only a minor component. We will revise the manuscript to clarify this distinction ("Although acetate occurs naturally, the use of pure acetate at the concentrations applied in this experiment does not reflect the chemical complexity of natural DOC pulses.").

**Comment 19:**
Figures: We cannot read the labels on any of the ordinations (i.e. Figure 2 and all the other ordinations in the supplementary) because they are stacked together or placed below the points. Also, some of the figures are in the text whereas others are not. It is confusing. From a reviewer's perspective, having the figure in the text is far better.

**Response to comment 19:**
Thank you for this comment. We will improve the figures to ensure the labels are readable. The best approach may be to decrease the font size and adjust the labels slightly. Regarding the position of the figures in the text, we agree that including them in the text makes more sense. In this case, the figures are automatically shifted to the end by the LaTeX template, even if they are inserted in the Results section. We apologize for the inconvenience and try to provide a better solution.